# Competition and Exploitation for Ecological Capital Embodied in International Trade: Evidence from China and Its Trade Partners

**Zhaohua Li [1], Zhiyun Zhu [1,\*] and Shilei Xu [2]**

1   School of Economics, Huazhong University of Science and Technology, Wuhan 430074, China; zhaohuali@hust.edu.cn
2   School of Economics, Nankai University, Tianjin 300071, China; xushilei3740@126.com
\*   Correspondence: melody_zhiyunzhu@163.com

**Abstract:** In order to identify ecological relationships of participating countries in the transfer system of ecological capital embodied in global trade, this paper takes the international trade of China and its partners from 2002 to 2017 as a case, and uses the ecological footprint (EF) measured by the optimized product land-use matrix (PLUM) method to quantify ecological capital for the value of macro-ecological resources, then uses the ecological network analysis (ENA) method to construct a complete transfer network of trade-embodied ecological capital and uses a utility analysis to identify ecological relationships between trading countries. Our results show that: (1) Throughout the study period, competition relationships with 61% dominated in the network, and the countries that have a pair-wise competition relationship with China are mainly located in central and western Europe, northeastern Europe, North America, southern Asia and eastern Asia. (2) Indirect utility determines the dominant ecological relationship in system, and it mainly converts dominant ecological relationships from control to competition by transforming exploit into competition. (3) China is looking to creating a more mutually beneficial trading environment at the expense of its own interests. (4) A global crisis event is likely to result in the control of ecological capital in more countries, and in its aftermath, the world is likely to be in a highly competitive environment. Reducing ecological capital consumption by improving energy efficiency and optimizing the global trading environment into a trading system dominated by mutualism relationships can be effective ways for countries around the world to achieve sustainable development post-COVID-19 crisis.

**Keywords:** trade-embodied ecological capital; ecological footprint; macro-ecological resource; ecological relationship; ecological network analysis; global crisis

## 1. Introduction

With increasingly accelerated development of economic globalization and trade liberalization, international trade has become the most important method of economic development. However, most scholars usually study the economic benefits brought by international trade, but ignore the transfer of ecological capital embodied in trade. Moreover, because of the increasing separation of production and consumption between countries, it is increasingly necessary to study trade-embodied ecological capital transfer [1,2].

Ecological capital, also known as ecological capital, is used to represent the value of ecological resources and has extensive research in the field of ecological economics [3–5]. Generally, researches on ecological capital embodied in international trade use two methods. The complex network method is used to study the direct transfer of ecological capital embodied in trade [1,6]. Since the behavior of the real global ecological capital transfer system is similar to that of an organism with a certain organizational structure and functional relationship, it can be analogized to an ecosystem [7,8]. However, in an ecosystem, indirect utility identified by indirect transfer often determines individual and system behavior [7,9,10]. Ecological network analysis (ENA) is an effective method to identify the

material and energy transfer information in a system, and it can quantitatively study the interaction between participating members in a network, so as to understand the integrity and complexity of the ecosystem [11–13]. Therefore, the ENA method is widely used to study the complete transfer of trade-embodied ecological capital, which takes both indirect and direct transfer into account. In recent years, ENA has mainly focused on the complete transfer of ecological capital for the value of a certain specific ecological resource or two specific ecological resources nexuses, such as $CO_2$ [14,15], virtual water [7], energy [16], energy-water nexus [9] and water-land nexus [17]. Existing studies show that the ENA method can reveal the complete transfer of ecological capital embodied in trade goods, not just the direct transfer at both ends of trade. However, previous literature only used one or two specific ecological resources to represent ecological capital, which indeed cannot explore the transfer of trade-embodied ecological capital more comprehensively and accurately. In order to better distinguish ecological resources of different material compositions, we define two concepts about ecological resources from both the micro and macro perspectives. Micro-ecological resource refers to any specific ecological resource of single material, and macro-ecological resource refers to aggregating of all micro-ecological resources, which can more comprehensively quantify the total amount of ecological capital. Therefore, based on our definition of ecological resource, the existing researches on ecological capital embodied in trade are limited to micro-ecological resource, and did not have research on trade-embodied ecological capital for the value of a macro-ecological resource.

Ecological footprint (EF) is an appropriate way to quantify the value of macro-ecological resources [18,19]. EF measures the regeneration capacity of the biosphere occupied by human activities. The EF of a country refers to the total area of land-use required to produce the goods it consumes, to absorb the wastes it produces, and to provide space for its infrastructure. EF is divided into six types of land-use: cropland, pasture, forest, marine, built-up land and carbon sink land. The input-output (IO) and product land-use coefficient (PLUC) are two main methods for calculating EF. As the IO method has problems such as the limitations of long-term data acquisition and low commodity resolution (IO resolution can only reach sectors or industries, and PLUC can be accurate to "orange" and "orange juice") and relatively complex coefficient matrix construction [20]. However, most of the existing studies using the PLUC method have problems such as the lack of standards for trade product selection, and scattered information sources for yield coefficients and equivalent factors [21–23]. D D Moran et al. [24] further extended the PLUC method to the product land use matrix (PLUM) method, but the matrix constructed cannot be directly calculated, and the matrix calculation formula was not listed. Therefore, in this article, we build a new framework for calculating EF based on the PLUM method. It is composed of the computable matrices including the products trade-volume vector, the land-use conversion coefficient matrix and the equivalent factor vector and their calculation formulas, which can address the problems existing in previous studies.

In the past few decades, the rapid growth of China's foreign trade has led to an increase in the exchanges of ecological capital between China and its trading partners. This has attracted more and more researchers' attention in recent years. Therefore, this article takes the trade of China and its partners as a case, uses the PLUM method to calculate the EF (that is the trade-embodied ecological capital for the value of a macro-ecological resource) in trade products; then uses the ENA method to build the complete transfer network of trade-embodied ecological capital, and uses utility analysis to identify the ecological relationships (control, exploitation, competition and mutualism) in the network between China and its trading partners and its changes over time.

This paper contributes to the current literature in four important aspects. First, we distinguished the ecological resource of different material composition by defining a micro-ecological resource and macro-ecological resource. Second, we used EF to quantify the ecological capital for the value of a macro-ecological resource, and used the optimized PLUM method to provide EF with a more calculable framework. Third, we incorporated the PLUM method into the ENA method. Fourth, this research provides scientific support

for clarifying the ecological responsibilities between trading countries and optimizing the global ecological capital transfer system.

The rest of this paper is organized as follows: Section 2 presents the research data and its sources, and the construction of the calculation model of EF and the trade-embodied ecological capital transfer network; Section 3 identifies and interprets the ecological relationships between countries and their changes over time; Section 4 discusses the insights evoked from our study; and Section 5 is our conclusion.

## 2. Data and Methods

### 2.1. Data (All the Datasets Used in This Paper Can Be Provided on Request)

We used the PLUM method to calculate the ecological footprint (EF) and used EF to quantify the transfer of trade-embodied ecological capital for the value of a macro-ecological resource (hereinafter, ecological capital for the value of a macro-ecological resource is referred to as ecological capital) between countries. This paper first selected 40 key trading countries with a total GDP accounting for 85% of the global total (world input-output database; http://www.wiod.org/, accessed on 1 August 2020). Since the Food and Agriculture Organization (FAO; http://www.fao.org/home/en/; accessed on 30 August 2020) lacks data from Taiwan of China, this paper takes 39 countries excluding Taiwan of China as the research object. The 39 countries specifically included China and its trading partners as follows: Austria, Australia, Brazil, Belgium, Bulgaria, Canada, Cyprus, Czechia, Germany, Denmark, Spain, Estonia, Finland, France, The United Kingdom, Greece, Hungary, India, Indonesia, Ireland, Italy, Japan, Lithuania, Luxembourg, Latvia, Malta, Mexico, The Netherlands, Poland, Portugal, Romania, The Russian Federation, Slovakia, Slovenia, Korea, Switzerland, Turkey, and The United States of America. All the country names in the following figures use the three-digital alpha by the International Standardization Organization (ISO3-digit Alpha).

When calculating the EF, in terms of the selection of traded goods, because built-up land accounts for less than 10% of the total land use in the global scale and accounts for less than 2% of the embodied ecological footprint of traded goods, it hardly affects the final result. Therefore, this study followed the approach proposed by D D MORAN et al. [24] and divided the calculation of EF into areas of cropland, pasture, forest, marine and carbon sink land, and calculated the EF of all goods imported and exported between 39 countries including China and its trading partners during the study period. In terms of commodity trade data sources, the trade data of agricultural products, livestock, animal husbandry products and fishery products were all sourced from UNComtrade (https://comtrade.un.org/data; accessed on 1 August 2020), using the HS02 commodity code; the trade data of forest products were all sourced from the Food and Agriculture Organization (FAO), using the FAO commodity code; the commodity trade data of carbon sink land came from UNComtrade, using SITC Rev.1 commodity code. When calculating the EF of carbon sink land, the embodied energy density and power energy coefficient of the goods were obtained from the National Footprint Accounts 2018 (NFA2018) in the Global Footprint Network (GFN; https://www.footprintnetwork.org/, accessed on 31 October 2019). However, since the NFA2018 version does not provide the world's carbon density of electricity and heat, this paper adopted the $CO_2$ accounting worksheet in the Hungarian account of the NFA2018 version to calculate the carbon density of the world's electricity and heat. All product yield coefficients and land-use equivalent factors required for the calculation of EF in this study were derived from the NFA2018 version. Since the latest NFA2018 version only counted data from years before 2018, the study period of this paper is 2002~2017.

### 2.2. Methods

This paper used 2014 as the base year and used global constant yield to quantify the transfer of the trade-embodied ecological capital between China and its partners from 2002 to 2017. That is, under the assumption that the ecological productivity of the world's land remains unchanged, we used the world land average yield coefficient of 2014 for every

year from 2002 to 2017. This makes the transfer of ecological capital comparable in space and time. Based on the calculated EF, we constructed an ecological capital transfer network between China and its trading partners, and then used utility analysis in the ENA method to study the ecological relationship and its temporal and spatial distribution among the network entities. The following are the specific method steps.

### 2.2.1. Calculation of EF Embodied in Trade Products

Step1: Trade Volume Vector (*V*)

$$V = [v_1 \ v_2 \ \cdots \ v_i \ \cdots \ v_n] \tag{1}$$

where $v_i$ denotes the trade volume of the *i*-th product, $n$ denotes the total number of traded goods, $i \in [1, n]$.

Step2: Product Land-Use Conversion Coefficient Matrix (*C*)

$$C = \begin{bmatrix} c_{1cr} & c_{1ps} & c_{1fr} & c_{1mr} & c_{1cs} \\ c_{2cr} & c_{2ps} & c_{2fr} & c_{2mr} & c_{2cs} \\ \vdots & \vdots & \vdots & \vdots & \vdots \\ c_{icr} & c_{ips} & c_{ifr} & c_{imr} & c_{ics} \\ \vdots & \vdots & \vdots & \vdots & \vdots \\ c_{ncr} & c_{nps} & c_{nfr} & c_{nmr} & c_{ncs} \end{bmatrix} \tag{2}$$

In Formula (2),

$$c_{it} = \frac{1}{y_{it}}, \ t \in \{cr, ps, fr, mr\} \tag{3}$$

$$c_{ics} = PEI_i * EPC_i * WECI_i * CFI_i \tag{4}$$

where *t* denotes land-use type, *cr, ps, fr, mr, cs* denote cropland, pasture, forest, marine and carbon sink land, respectively; $c_{it}$ denotes the *t*-th type land-use conversion coefficient of the *i*-th product, and $y_{it}$ denotes the *t*-th type land-use yield coefficient of the *i*-th product (when calculating the EF of cropland and pasture of livestock and animal husbandry products, the land-use conversion coefficients of the products are respectively equal to cropland density and pasture density); $c_{ics}$ denotes the carbon sink land-use conversion coefficient of the *i*-th product, and $PEI_i$, $EPC_i$, $WECI_i$ and $CFI_i$ denote the energy density, power energy coefficient, world electricity and heat energy carbon density and carbon footprint density of the *i*-th product, respectively.

Step3: Equivalent Factor Vector *E*

$$E = \begin{bmatrix} e_{cr} \\ e_{ps} \\ e_{fr} \\ e_{mr} \\ e_{cs} \end{bmatrix} \tag{5}$$

where $e_{cr}$, $e_{ps}$, $e_{fr}$, $e_{mr}$ and $e_{cs}$ denote the equivalent factors of cropland, pasture, forest, marine and carbon sink land, respectively. By multiplying various types of land-use area by corresponding equivalent factors, they can be converted into areas with the same ecological productivity per unit area, so that the area of different types of land-use is comparable.

Step4: Ecological Footprint ($EF_{ij}$)

$$EF_{ij} = VCE = \begin{bmatrix} ef_{cr} & ef_{ps} & ef_{fr} & ef_{mr} & ef_{cs} \end{bmatrix} \begin{bmatrix} e_{cr} \\ e_{ps} \\ e_{fr} \\ e_{mr} \\ e_{cs} \end{bmatrix}$$

$$= \sum_{t=cr}^{cs} ef_t * e_t, \ t \in \{cr, ps, fr, mr, cs\}$$

(6)

where $EF_{ij}$ denotes the embodied ecological capital transferred from country $j$ to country $i$ through trade; $V$, $C$ and $E$ denote the trade volume vector, product land-use conversion coefficient matrix and equivalent factor vector, respectively; $ef_t$ denotes the total EF of the $t$-th category land embodied in the $n$ products imported from country $i$ to country $j$, that is, $ef_t = \sum_{i=1}^{n} v_i * c_{it}$.

### 2.2.2. Ecological Network Analysis
Construction of Ecological Network Model

Based on the above calculation of the EF between 39 countries, we constructed the trade-embodied ecological capital transfer network model. The countries are the network nodes, the flow of ecological capital embodied in trade between countries is the path between nodes, and the amount of ecological capital flow ($EF_{ij}$) is the path width. Paths in the network mean values of the transfer of ecological capital is embodied in traded goods. We assumed that the remaining countries and/or economies in the world are part of the external environment, and the 39 countries studied in this article constitute an integrated transfer system of ecological capital. Figure 1 shows the trade-embodied ecological capital transfer network model of 39 key trading countries and/or economies in 2017.

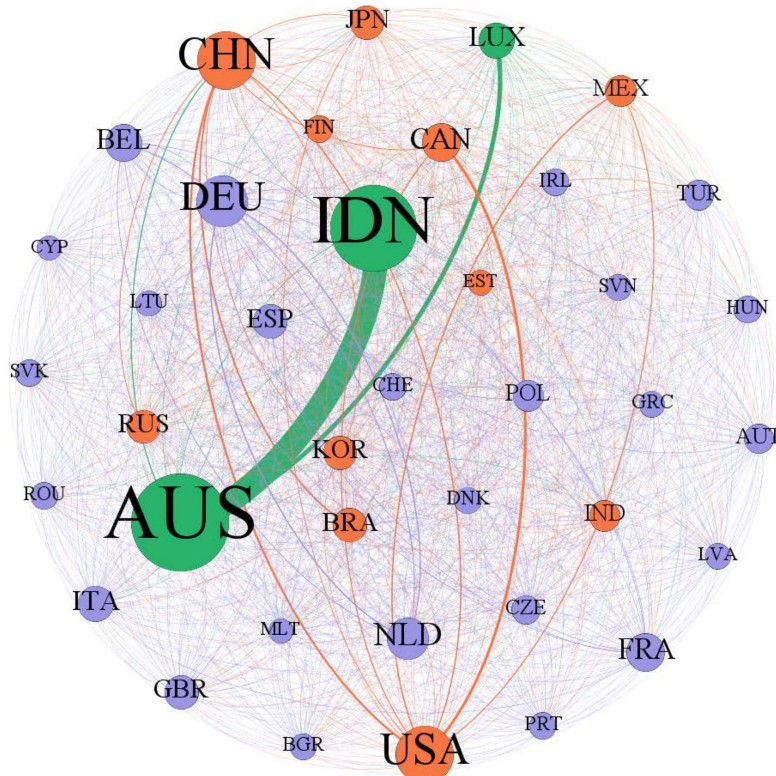

**Figure 1.** The transfer network of trade-embodied ecological capital between China and its trading partners.

A node represents a country, the larger the node, the stronger its corresponding country's ecological capital transfer capacity; an edge represents an ecological capital transfer path, the thicker an edge, the greater the amount of ecological capital transfer between countries; the clockwise direction is the ecological capital transfer direction.

Ecological Relationships

By utilizing the utility analysis of the ecological network analysis (ENA), we can quantify the functional interrelationships among various nodes, which resemble the four major ecological relationships in a natural ecosystem: competition, exploitation, control and mutualism. In the ecological network model, the material-balance principle must be observed, that is, the system input is equal to the system output. The specific formula is as follows:

$$\sum EF_{ij} + z_i + S = \sum EF_{ji} + y_i \tag{7}$$

where $EF_{ij}$ denotes the direct ecological capital transfer from node $j$ to node $i$, $z_i$ denotes the environmental input obtained by node $i$, and $S$ denotes the ecological capital storage of node $i$; $EF_{ji}$ denotes the direct ecological capital transfer from node $i$ to node $j$, $y_i$ denotes the environment output of node $i$.

We define $T_i$ as the sum of the flows into node $i$, including cross-boundary inputs from the environment into country $i$. The specific calculation formula is as follows:

$$T_i = \sum_{j=1}^{n} E_{ij} + z_i \tag{8}$$

where $z_i$ is the cross-boundary inputs to country $i$ from the rest of the countries and/or economies in the world that excluded the 39 selected countries and/or economies.

Based on the above indicators, we can calculate a dimensionless direct utility intensity matrix ($D$), where $d_{ij}$ is the element of matrix $D$, which denotes the direct utility of the trade-embodied ecological capital from country $j$ to country $i$. The specific formula is as follows:

$$D = [d_{ij}] = \left[\frac{E_{ij} - E_{ji}}{T_i}\right] \tag{9}$$

According to the matrix $D$, a dimensionless integral utility intensity matrix $U$ is computed as follows [10]:

$$U = [u_{ij}] = D^0 + D^1 + D^2 + D^3 + \cdots + D^m + \cdots = (I - D)^{-1} \tag{10}$$

where $U$ is integrated utility intensity among nodes, indicating the integrated relationships between countries in the network. The matrix $D^0$ is the self-feedback of the flows within each node, which is not taken into consideration since we focus on the flows between nodes. $D^1$ is the direct flow utility that pass along the pathways of length 1. $D^2$ is the indirect flow utility that passes along the pathways of length 2, which means it passes a third country. $D^m$ ($m > 2$) represents the indirect flow utilities along the paths of $m$ steps. $I$ is the identity matrix, and $u_{ij}$ is the dimensionless integral utility value of $d_{ij}$.

According to matrix $U$ and matrix $D$, we can respectively obtain two sign matrices $Sign(U)$ and $Sign(D)$, $su_{ij}$ and $sd_{ij}$ are their constituent elements respectively, and the signs determine the ecological relationships between nodes. Taking $Sign(U)$ as an instance, if $(su_{ji}, su_{ij}) = (+, -)$, country $j$ obtains net ecological capital from country $i$; that is, country $j$ benefits from trade with country $i$. Country $j$ exploits country $i$. If $(su_{ji}, su_{ij}) = (-, +)$, country $j$ is exploited by country $i$; that is country $i$ controls country $j$. If $(su_{ji}, su_{ij}) = (-, -)$, country $i$ and country $j$ both obtain net ecological capital from other countries; that is, country $j$ and country $i$ are in a competition relationship. If $(su_{ji}, su_{ij}) = (+, +)$, country $j$ and country $i$ benefit from each other and come to a win-win situation in an ecological capital network of international trade.

From a system perspective, the network mutualism index (*NMI*) and Synergism Index (*SI*) are used to quantify the integrated ecological relationship shown by the whole network system [9,25,26]. *NMI* can summarize the proportion of mutualism relationships in the entire ecological network. *SI* evaluates the mutualism ability of the entire ecological network, that is, when *NMI* > 1 and *SI* > 0, the positive utility dominates the network, that is to say, the system is mutualistic. Conversely, if *NMI* < 1 and *SI* < 0, the competitive relationship dominates the network, that is, the system is competitive. Otherwise, the dominant ecological relationship of the system is uncertain. The specific calculation formula of *NMI* and *SI* is as follows:

$$NMI_{year} = SignU(+)_{year} / SignU(-)_{year} \tag{11}$$

$$SI_{year} = \sum_{i=1}^{n} \sum_{j=1}^{n} u_{ij} \tag{12}$$

$$TNMI = \sum_{year=2002}^{2017} SignU(+)_{year} / \sum_{year=2002}^{2017} SignU(-)_{year} \tag{13}$$

$$TSI = \sum_{year=2002}^{2017} SI_{year} \tag{14}$$

where $NMI_{year}$ denotes the network mutualism index of the year, $year \in [2002, 2017]$, $year \in N^+$; $SignU(+)_{year}$ and $SignU(-)_{year}$ respectively denote the total number of positive signs and negative signs in the sign matrix of the matrix $U$ of the current year; $SI_{year}$ denotes the synergism index of the year; *TNMI* and *TSI* denote the total network mutualism index and synergism index throughout the study period.

## 3. Results and Analysis

### 3.1. Integrated Changes in Ecological Relationships

This paper used the trade between China and its trading partners as a case to study the changes in the ecological relationship between countries over time. Based on the calculation of the trade-embodied ecological footprint (EF), we constructed the trade-embodied ecological capital transfer network from 2002 to 2017, and then through the utility analysis of the ENA method, we determined the ecological relationship (exploitation, control, competition and mutualism) between China and its trading partners and its changes over time.

During the entire study period, a total of 608 pairs of relationships occurred between China and its trading partners. According to Figure 2, in terms of the proportion of overall ecological relationships, control/exploited relationships, competition relationships and mutualism relationships account for 36%, 61% and 3% of the total, respectively, with the competition relationship accounting for the highest percentage. In terms of the proportion of ecological relationships per year, it can be seen that the proportion of ecological relationships between China and its trading partners changed dramatically before and after the 2008 global financial crisis, i.e., the proportion of control relationships surged from 13.16% in 2007 to 39.47% when the global financial crisis occurred in 2008, while the other three ecological relationships all declined to varying degrees; after the occurrence of the global financial crisis in 2009, a proportion of control relationships plummeted to the 2007 level, while a proportion of competition relationships surged to 71.05%. This suggests that a global crisis event is likely to result in the control of ecological capital in more countries, and that in its aftermath the world is likely to be in a highly competitive environment, which is likely to be the result of robust economic recovery plans in each country. Because global crisis events lead to a new global political and economic equilibrium, each country must compete to defend its interests in this new equilibrium. For China, the proportion of exploited relationships has increased since 2016, while control relationships have decreased; competition relationships with its trading partners are slowly weakening and mutualism relationships are gradually increasing. This suggests that China is looking to create a more mutually beneficial environment at the expense of its own interests.

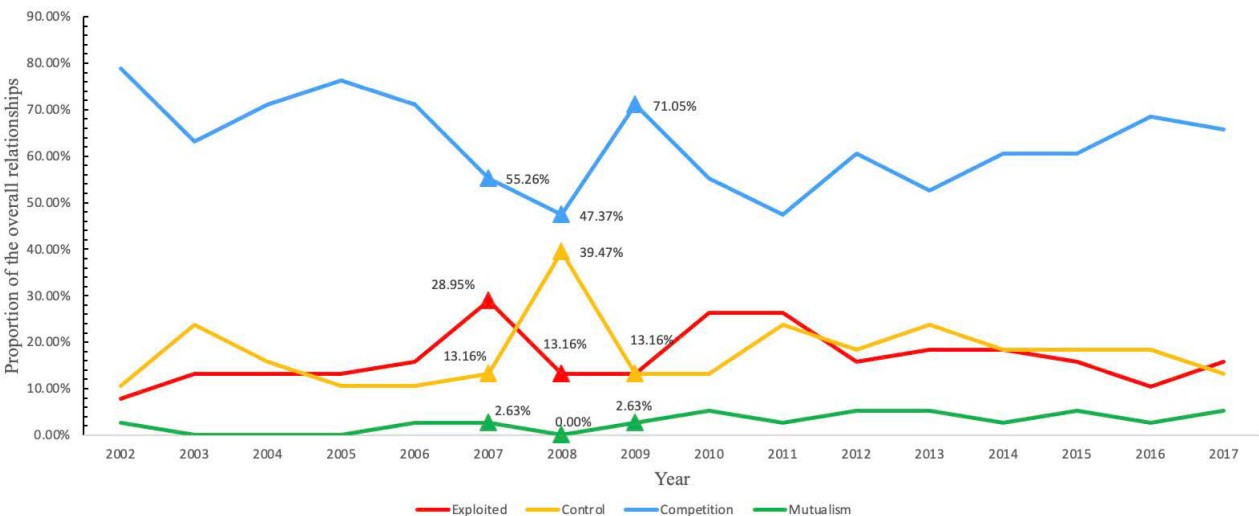

**Figure 2.** Changes in ecological relationship proportion during the research period.

These four ecological relationships seem to have some agglomeration characteristics geographically. Mapping the ecological relationship between China and 38 other key trading countries to the geographical location, we can find that in the 16 years of the study period, 82% of the countries that equipped the pair-wise competition relationship with China are located in Europe and about 16% are located in Asia; 50% of the countries over which China has control are located in Europe and about 20% are located in the Americas; 79% of the countries that have an exploitative effect on China are located in Europe and about 16% are located in the Americas; only Europe had a mutualism relationship with China. It shows that in the trade-embodied ecological capital transfer system of China and its trading partners, the four types of ecological relationships mainly occur in Europe or involve Asia or America.

NMI(U) and SI(U) represent the network mutualism index and synergism index, respectively, determined by the complete utility caused by complete transfer; NMI(D) and SI(D) represent network mutualism index and synergism index, respectively, determined only by direct utility caused by direct transfer.

From a system perspective, this paper uses the *NMI* and *SI* indexes to quantify the dominant ecological relationship in the trade-embodied ecological capital transfer system of China and its trading partners during the whole study period. Observing Figure 3, we can find that the NMI(D) determined only by direct utility is always equal to 1 during the study period, it is impossible to judge the dominant ecological relationship of this situation. At the same time, we found that during the study period, NMI(U) is always less than 1, and SI(U) is always less than 0, that is, the system determined by the complete utility presents a competitive-led ecological environment every year. NMI(U) fluctuates relatively smoothly, and SI(U) fluctuates greatly, but in recent years there has been an upward trend. It shows that the competition relationship in the trade-embodied ecological capital transfer system of China and its trading partners is gradually improving. All in all, TNMI < 1 and TSI < 0, which indicates that the system of China and its 38 trading partners is dominated by competition throughout the study period.

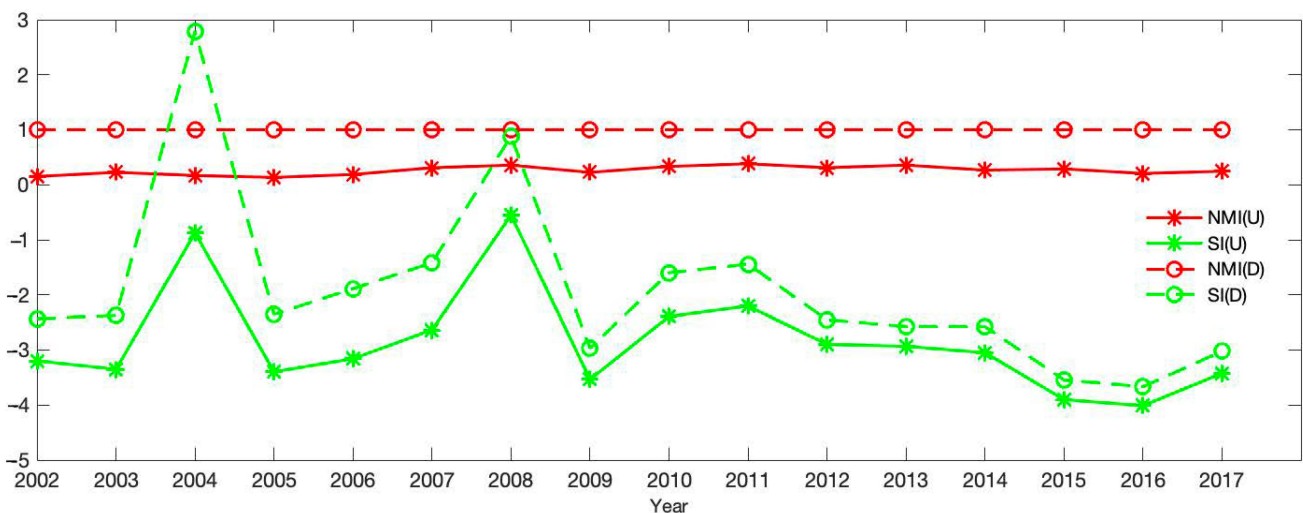

**Figure 3.** The trade-embodied ecological capital transfer system fitness index of China and its trading partners.

### 3.2. Changes of the Ecological Relationship between China and its Trading Partners

According to the utility analysis of the ENA method, there are four ecological relationships in the trade-embodied ecological capital transfer system of China and its trading partners. For China, the names and symbols of these four ecological relationships are: control (−, +), exploited (+, −), competition (−, −) and mutualism (+, +). In order to study the impact of indirect utility on the ecological environment of the system, we compared and analyzed the difference between the ecological relationship of countries identified by sign (D) (the sign matrix determined by the direct utility intensity matrix D) and sign (U) (the sign matrix determined by the complete utility intensity matrix U including direct utility and indirect utility). Figures 4 and 5, respectively, reveal the paired ecological relationship between China and its trading partners as determined by sign (D) and sign (U) during the study period. In this article, we believe that among all the ecological relationships formed between any two countries during the entire study period, the ecological relationship that occupies the highest proportion is the deterministic ecological relationship comprehensively displayed by these two countries during the whole research period (when the different ecological relations formed by two countries during the study period have the same highest proportion, the ecological relationship with the highest proportion in the last year is considered to be the deterministic ecological relationship formed by the two countries).

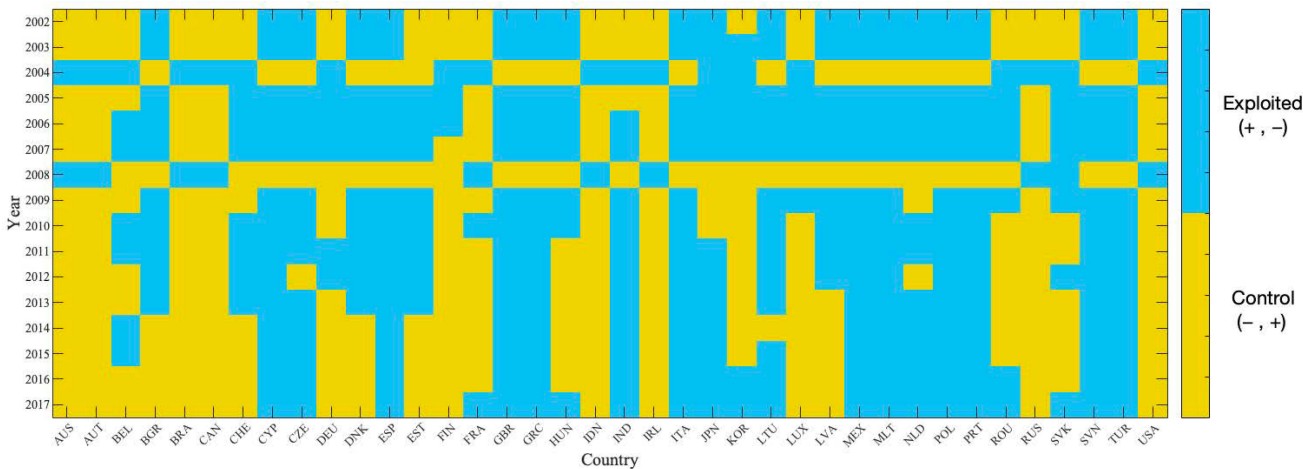

**Figure 4.** Pair-wise direct ecological relationship between China and its trade partners from 2002 to 2017.

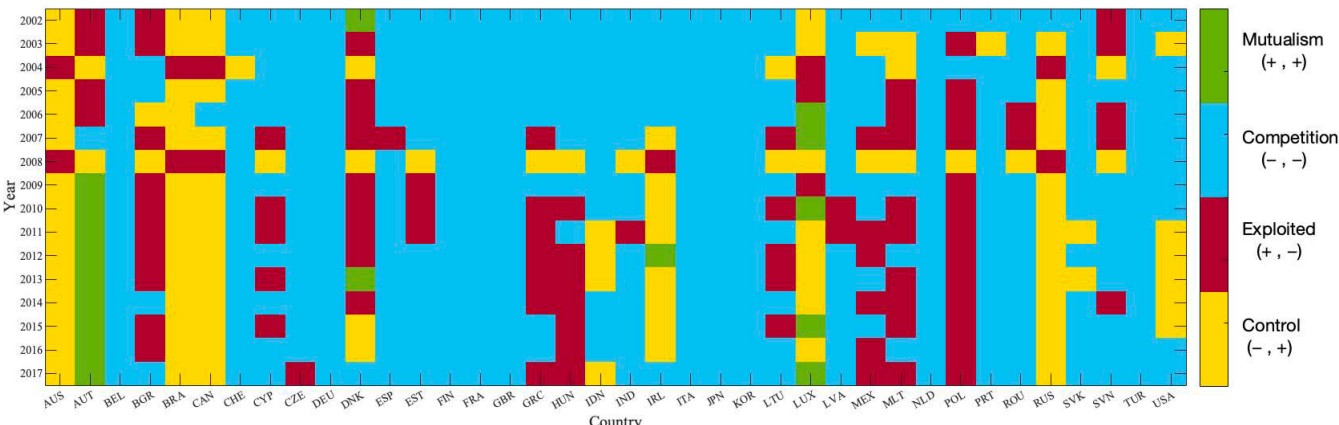

**Figure 5.** Pair-wise integral ecological relationship between China and its trade partners from 2002 to 2017.

Based on the definition principle of the deterministic ecological relationship we provided above, the ecological relationship between countries determined by direct utility and complete utility can be determined. Figure 4 represents the ecological relationship between countries determined by sign (D). It can be found that the ecological relationship between countries determined by direct utility only includes the control and exploited relationship, in which they account for 55% and 45% of the total, respectively. It shows that only under the influence of direct utility, the trade-embodied ecological capital transfer system of China and its trading partners is dominated by a control relationship. Figure 5 represents the ecological relationship between countries determined by sign (U). It can be found that the ecological relationship between countries determined by the complete utility includes all possible relationships, namely control/exploitation, competition, and a mutualism relationship, which accounts for 36%, 61%, and 3% of the total, respectively. It shows that under the effect of complete utility, the trade-embodied ecological capital transfer system of China and its trading partners is dominated by competition relationships. By comparing the results obtained in Figures 4 and 5, we can find that there are many differences, that is, not only that the types of ecological relations are different, but also the dominant ecological relations in the system. That is to say, the existence of indirect utility changes the ecological relationship determined by direct utility between countries and the dominant ecological relationship of the entire system, that is, indirect utility has a decisive effect on system behavior.

The horizontal axis denotes the country, and the vertical axis denotes the year; the two colors in the legend on the right denote the two ecological relationships between China and another country. For example, the color of the square of AUS (the first country on horizontal axis) in the figure in 2008 is blue, which indicates that China is being exploited by AUS, which is an exploited relationship.

The horizontal axis denotes the country, and the vertical axis denotes the year; the four colors in the legend on the right denote the four ecological relationships between China and another country. For example, the square color of AUT (the second country on horizontal axis) in the figure in 2007 is blue, which indicates that there is a pair-wise competition relationship between China and AUT.

In order to further study how indirect utility determines system behavior, we analyzed the ecological relationship determined by complete utility and its transformation during the entire study period. Figure 6 reveals the transformation of ecological relationships each year during the study period. It can be found that there are six mutual transformations of ecological relationships throughout the study period, namely, control transforms into exploited, control transforms into competition, control transforms into mutualism, exploited transforms into control, exploited transforms into competition and exploited transforms into mutualism. In addition, the frequency of the transformation of ecological relationships that exist every year is different, in which the transformation from exploited to competition

is the most critical, and the control is the relationship that transforms into mutualism the most. It shows that in the trade-embodied ecological capital transfer system of China and its partners, indirect utility is mainly to transform the dominant ecological relationship in the system from control to competition by converting the exploited relationship into the competition relationship, and make China reduce its own interests to contribute to the mutually beneficial ecological environment of system.

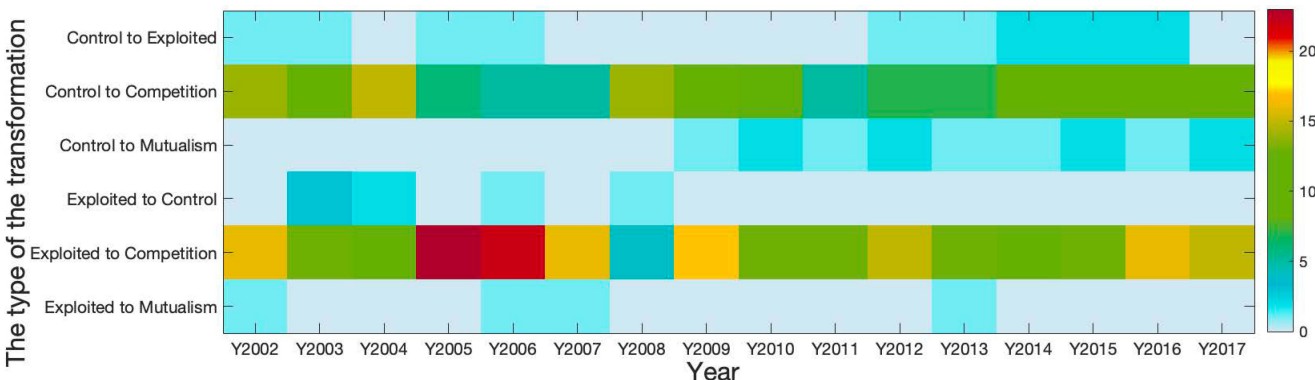

**Figure 6.** The transformation of ecological relationships determined by complete utility during the whole study period.

The horizontal axis denotes the year, and the vertical axis denotes the type of transformation of ecological relationships; the legend on right goes from bottom to top, from lighter to darker colors, i.e., the darker the color, the more frequent the transformation of that type of ecological relationship.

Based on the above-identified ecological relationships between countries, the ecological responsibilities of each country can be clarified, that is, countries that consume the ecological capital of other countries should bear corresponding ecological responsibilities. Figures 5 and 7 respectively reveal the ecological relationship and geographic distribution between China and its trading partners during the entire study period. According to Figure 5, it can be determined that China has a controlling effect on the ecological capital of six other countries including Australia, Brazil, Canada, Ireland, Luxembourg, and Russia; four countries including Bulgaria, Denmark, Malta and Poland are exploiting China's ecological capital; Austria is the only country that has mutualism relationship with China. A total of 27 countries including Belgium, Switzerland, Cyprus, The Czech Republic, Germany, Spain, Estonia, Finland, France, The United Kingdom, Greece, Hungary, Indonesia, India, Italy, Japan, Korea, Lithuania, Latvia, Mexico, The Netherlands, Portugal, Romania, Slovakia, Slovenia, Turkey and The United States have a pair-wise competition relationship with China, and it is they that prompted the trade-embodied ecological capital transfer system of China and its trading partners to be dominated by competition. According to Figure 7, it can be found that the countries that equipped a pair-wise competition relationship with China are geographically concentrated in central and western Europe, northeastern Europe, North America, southern Asia and eastern Asia.

The red line, green line, yellow line and blue line, respectively, represent the distribution of countries that equipped pair-wise relationships of competition, control, exploit and mutualism with China.

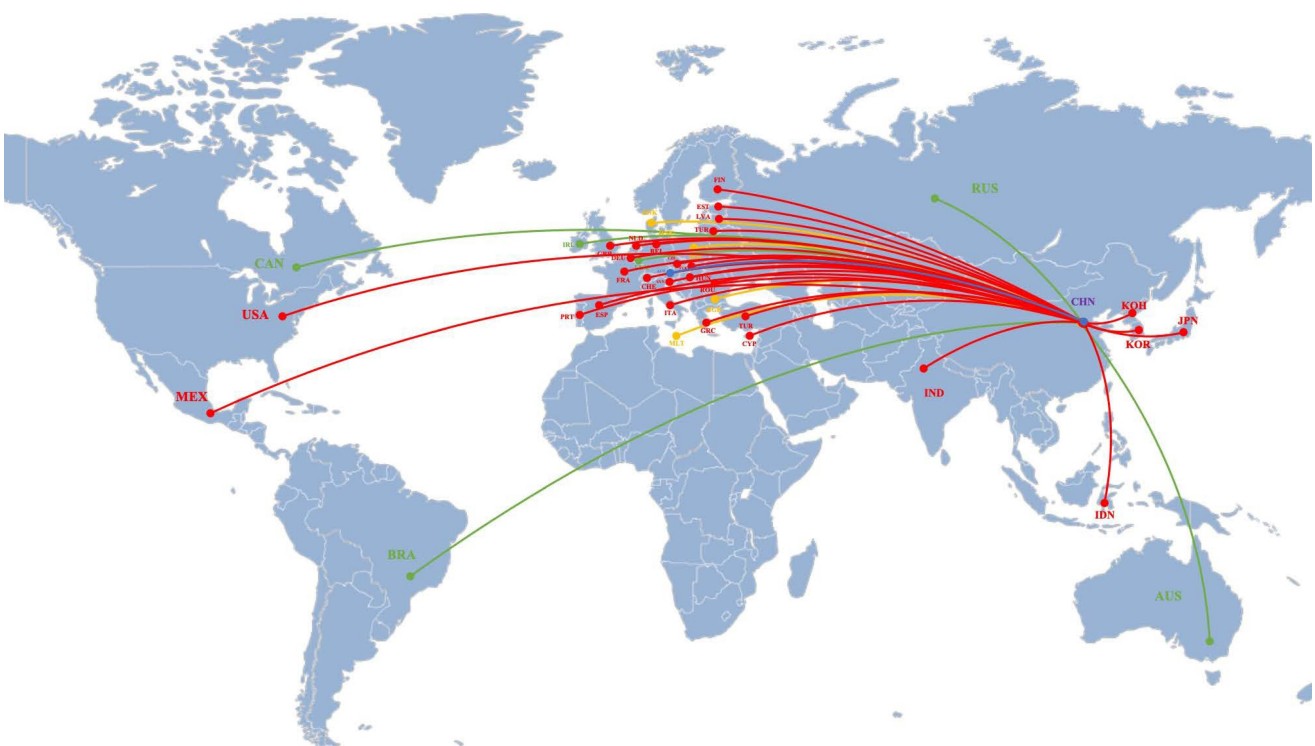

**Figure 7.** Geographical distribution of the ecological relationship between China and its trading partners.

## 4. Discussion

In this study, we used the PLUM method to calculate the ecological footprint (EF), quantify the complete transfer of ecological capital in trade products with EF, and use the utility analysis of the ENA method to identify the ecological relationship between China and its trading partners and their changes over time. The following are different insights evoked from our results.

Firstly, we focused on the complete transfer of ecological capital embodied in international trade by using the ENA method, while previous studies mostly used the input-output (IO) method to calculate the direct transfer of ecological capital embodied in trade. For example, C DUAN AND B CHEN [9] used the IO method to measure the direct transfer of virtual water embodied in energy trade products. Zhang et al. used the IO methods to calculate the direct transfer of embodied $CO_2$ [8,14,15]. However, this paper not only verifies the direct transfer, but also studies how indirect utility identified by indirect transfer changes system behavior. In the trade-embodied ecological capital transfer system of China and its partners, we found that the ecological relationships determined only by direct utility only have the control/exploitation relationship, and the control relationship dominates at 55%. Due to the influence of indirect utility, ecological relationships such as control/exploitation, competition, and mutualism actually exist in the system, and the competition relationship is dominant at 61%. Moreover, indirect utility mainly makes the system dominated by the competition relationship by transforming the exploited relationship into the competition relationship, and makes China reduce its own interests to contribute to the mutually beneficial ecological environment of the system. At this point, the ecological environment of those countries whose ecological capital is exploited is affected by varying degrees with the ongoing trade. Therefore, on the premise of clarifying different ecological relationships between individual countries, and in order to reduce the impact of trade on individual countries, the world should work towards the establishment of a binding protocol or regional ecological capital protection association, which considers the global ecological environment as a whole, in which each trading country should take responsibility for global ecological protection.

Secondly, indirect utility is mainly to transform the dominant ecological relationship in the system from control to competition by converting the exploited relationships into competition relationships, and make China reduce its own interests to contribute to the mutually beneficial ecological environment of the system. For China, a proportion of exploited relationships has shown an upward trend since 2016, while the proportion of control relationships has shown a downward trend. It shows that China is looking to create a more mutually beneficial trading environment at the expense of its own interests. In this regard, countries with a large number of controlling relationships, as the main beneficiaries of international trade, are expected to assume leadership, which is to make efforts to guide global countries to behave in an eco-friendly manner. In order to make the trade-embodied ecological capital transfer system between China and its partners finally present an eco-logical environment dominated by mutualism relations, in addition to trying to reduce competition relations, it is also necessary to control the generation of exploited relations and reduce the chances of the exploited relations turning into competition relations. That is to say, it is necessary to promote the transformation of the competitive relationship into the other three relationships.

Thirdly, the dominant ecological relationships of socio-economic systems of ecological resources of different material composition are different. Most of the existing researches focus on the study of micro-ecological resources, and believe that the dominant ecological relationship is mainly dominated by exploitation/control or competition relationships [27, 28]. For instance, at the level of virtual water transfer, it is mostly dominated by the exploitation/control relationship. X Mao and Z Yang [29] observed that the entire Baiyangdian Basin is dominated by 70% exploitation/control relationships; D Fang and B Chen [30] found that the Heihe River Basin is dominated by 67.7% exploitation/control relationships. At the level of $CO_2$ transfer, the dominant position between competition and exploitation/control relationships is almost equal. For example, Y Zhang et al. [8] found that competition and exploitation/control relations both dominate the $CO_2$ transfer network with a ratio of more than 40%, and mutualism accounts for the smallest proportion with 4%. When considering the level of ecological capital for the value of macro-ecological resource, our results show that in the trade-embodied ecological capital transfer network of China and its trading partners, competition occupies the dominant position with 61%, the proportion of exploitation/control is 36%, and mutualism occupies the smallest proportion with 3% of the complete relationship. That is to say, the dominant ecological relations and proportions in the socio-economic system of ecological resource transfer of different material composition are different, and compared with a micro-ecological resource, the competition in the transfer systems of a macro-ecological resource is more intense. In this regard, in order to effectively reduce competition for trade-embodied ecological capital on a global scale, the world should aim to encourage participatory alliances and try to match the final consumption of ecological capital with the corresponding ecological responsibility, which may be an opportunity for mutual compromise between countries with different ecological relationships. By participating in alliances that effectively avoid geopolitics and face fewer policy impediments, the formation of an optimal global framework for ecological protection is ultimately possible. Given that transfer of trade-embodied ecological capital is already inextricably linked to national/regional economic development and environmental protection [31], we suggest that national trade policies should be considered in conjunction with environmental policies.

Fourthly, the ecological relationship (control/exploitation, competition and mutu-alism) in the trade-embodied ecological capital transfer system between China and its partners has obvious geographical distribution characteristics. For example, Y Zhang et al. [8] found that more than 80% of competition and 75% of exploitation/control re-lationships occurred in Europe. According to the research results of this paper, 82% of the competition, 50% of the control, and 79% of the exploitation occurred in Europe or involved Asia or the Americas. More specifically, most of the countries that equipped the pair-wise control/exploitation relationship with China belong to Europe; the countries

that equipped the pair-wise competition relationship with China are mainly concentrated in central and western Europe, northeastern Europe, North America, southern Asia and Eastern Asia; the only country that has a mutualism relationship with China is Austria in Central Europe. Austria has a special geographical location as a rail transit hub in all directions across Europe. For China, it can connect Chinese industry with the European economic center and the world market; for Austria, under the Belt and Road initiative, whether it's the northern route via Russia or the southern route via Turkey and Iran is crucial. Therefore, based on the long-standing mutualism ecological relationship between China and Austria, under the framework of the Belt and Road initiative, China and Austria have great potential for cooperation.

Fifthly, a global crisis event is likely to result in the control of ecological capital in more countries, and in its aftermath the world is likely to be in a highly competitive environment, which is likely to be the result of robust economic recovery plans in each country. Therefore, in the event of a global crisis, countries around the world should prevent other countries from controlling their ecological capital by trading while ensuring national security; in the aftermath of a global crisis, countries should develop appropriate economic recovery plans and policies to increase the rate of economic development and overcome the intense competition that ensues. For example, with the COVID-19 global crisis that began in 2019 and continues to this day, the primary concern of global countries should now be how to reduce global competition while addressing the conditions of health and their own economic growth rates, thus contributing to global sustainability. At this point, reducing ecological capital consumption by improving energy efficiency and optimizing the global trading environment into a trading system dominated by mutualism relationships can be effective ways for global countries to achieve sustainable development in the aftermath of the COVID-19 global crisis.

## 5. Conclusions

This paper took the trade of China and its partners as a case, distinguished ecological resources of different material composition by defining micro-ecological resource and macro-ecological resource, used ecological footprint (EF) to quantify the ecological capital for the value of a macro-ecological resource, and used the optimized PLUM method to calculate the EF; then used the ENA method to construct the complete transfer network of ecological capital between China and its trading partners, and used utility analysis to more comprehensively and accurately identify the ecological relationship between countries and their changes over time. This research can provide scientific support for clarifying the ecological relationship and ecological responsibilities between countries and optimize the global ecological capital transfer system.

We identified 608 pairs of ecological relationships between China and its trading partners from 2002 to 2017. The trade-embodied ecological capital transfer system of China and its trading partners is dominated by competition with the proportion of 61%, and 82% of competition, 50% of control, and 79% of exploited relationships which occurred in Europe or involved Europe's shift to Asia or America. Indirect utility is mainly to transform the dominant ecological relationship in the system from control to competition by converting the exploited relationships into competition relationships. China is looking to create a more mutually beneficial trading environment at the expense of its own interests. China has a controlling role in the ecological capital of six other countries including Australia, Brazil, Canada, Ireland, Luxembourg, and Russia. At this time, China should be required to share part of the ecological responsibilities of these six countries. China's ecological capital has been exploited by countries including Bulgaria, Denmark, Malta and Poland. For this instance, China should ask these countries for compensation for the consumption of ecological capital. Austria is the only country that has a long-term and stable mutualism relationship with China. Therefore, China should actively formulate relevant policies for further friendly exchanges with Austria, and strive to develop Austria into a hub of foreign trade under the development framework of the Belt and Road initiative. The remaining

27 of the 38 countries formed pair-wise competition relationships with China. Therefore, in order to more effectively develop binding agreements or establish regional ecological capital conservation associations, mutual compromise and participation among competing countries in the system will be key to global eco-environmental optimization. Moreover, reducing ecological capital consumption by improving energy efficiency and optimizing the global trading environment into a trading system dominated by a mutualism relationship can be effective ways for countries around the world to achieve sustainable development post-COVID-19 crisis.

**Author Contributions:** Resources, Validation, Supervision, Project administration, Funding acquisition, Writing–Review & Editing, Z.L.; Conceptualization, Investigation, Formal analysis, Visualization, Writing–Original Draft, Z.Z.; Methodology, Software, Data Curation, S.X. All authors have read and agreed to the published version of the manuscript.

**Funding:** This research received no external funding.

**Institutional Review Board Statement:** Not applicable.

**Informed Consent Statement:** Not applicable.

**Data Availability Statement:** Global Footprint Network (https://www.footprintnetwork.org/ accessed on 31 October 2019); World Input-Output Database (http://www.wiod.org/ accessed on 1 August 2020); UNComtrade (https://comtrade.un.org/data accessed on 1 August 2020); Food and Agriculture Organization (http://www.fao.org/home/en/ accessed on 30 August 2020).

**Conflicts of Interest:** The authors declare no conflict of interest.

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
