# Peer review of "Competition and Exploitation for Ecological Capital Embodied in International Trade: Evidence from China and Its Trade Partners"

_sustainability, doi:10.3390/su131810020_

Round 1

Reviewer 1 Report

I would like to see the following improvements in the manuscript before making a concrete decision:

  1. The authors provide references for well-known facts or realities such as

‘Nowadays, with increasingly accelerated development of economic globalization 25 and trade liberalization, international trade has become the most important way of eco- 26 nomic development (Zhu et al., 2020).’

This wrong way of referencing should be eliminated from the manuscript. There are some other similar statements in the same way in the manuscript.

This paper first selects 40 key trading countries with a 116 total GDP accounting for 85% of the global total (Zhang et al., 2019; Zhang et al., 2017).  The authors should provide  primary data sources of this fact rather than relying on secondary references.

  1. The authors should avoid multiple references. To this extent, no more than 3 references in chronological order in a sentence are appropriate.
  2. The authors should provide a footnote that they are willing to share their data set in Excel format with those who wish to replicate the results of this research.
  3. The coloured figures of 4, 5 and 6 should be substituted with some other forms of graphs as they are difficult to follow.
  4. The authors should highlight the shortcomings of the adopted estimation methodology.

Reviewer 2 Report

Dear Authors,

Please find below and attached my comments and suggestions for your work.

Good luck!

Kind regards,

The Reviewer

Review Report Form

Manuscript ID: sustainability-1351535

Type: Article

Number of Pages: 26

Title: Competition and Exploitation for Ecological Capital Embodied in International Trade: Evidence from China and its Trade Partners

Authors: Zhaohua Li , Zhiyun Zhu * , Shilei Xu

 Date: 06 August 2021

Dear Authors,

I have carefully analyzed your article entitled “Competition and Exploitation for Ecological Capital Embodied in International Trade: Evidence from China and its Trade Partners”.

Congratulations for your work and valuable insights reflected in the content of the manuscript!

The structure of my Review Report Form takes into consideration two sections, namely: (A.) General overview of the article and strong points; and (B) Suggestions meant to improve your current manuscript.

(A.) General overview of the article and strong points:

  • General aim of the study and research methods used: The authors state that, in order to identify ecological relationships of participating countries in transfer system of ecological capital embodied in global trade, this paper takes international trade of China and its partners from 2002 to 2017 as a case, uses Ecological Footprint (EF) measured by optimized Product Land-Use Matrix (PLUM) method to quantify ecological capital for value of macro-ecological resources, then uses ecological network analysis (ENA) method to construct complete transfer network of trade-embodied ecological capital and uses utility analysis to identify ecological relationship between trading countries.
  • General results: The results of this study show that: (a) Throughout the study period, competition relationship with 61% dominated in the network, and the countries that have pair-wise competition relationship with China are mainly located in central and western Europe, northeastern Europe, North America, southern Asia and eastern Asia; (b) Indirect utility determines dominant ecological relationship in system, and it mainly converts dominant ecological relationship from control to competition by transforming exploited into competition; and (c) China is looking to creating a more mutually beneficial trading environment at the expense of its own interests.
  • Novelty of the study: This research provides a calculation framework for quantifying complete transfer of trade-embodied ecological capital, and provides scientific support for clarifying ecological responsibilities between trading countries and optimizing global ecological capital transfer system by identifying ecological relationship between trading countries comprehensively and accurately.

(B) Suggestions meant to improve your current manuscript:

Distinguished Authors I would kindly like to suggest inserting in your article a few ideas concerning the correlation between the effects of the Covid-19 global crisis, ecological capital, competitiveness, sustainability, and sustainable development, since these are key focuses these days. In this context, I had the chance to read a few interesting articles recently, among which I would like to mention: An Exploratory Study Based on a Questionnaire Concerning Green and Sustainable Finance, Corporate Social Responsibility, and Performance: Evidence from the Romanian Business Environment. J. Risk Financial Manag. 2019, 12, 162. DOI: 10.3390/jrfm12040162, link: https://www.mdpi.com/1911-8074/12/4/162.

Dear Authors, congratulations once again for your work and valuable insights reflected in the content of the manuscript, and I hope my comments will be of value to you!

Kind regards,

The Reviewer

Round 2

Reviewer 1 Report

Accept as is